# AlF–AlF Reaction Dynamics between 200 K and 1000 K: Reaction Mechanisms and Intermediate Complex Characterization

**DOI:** 10.3390/molecules29010222

**Published:** 2023-12-31

**Authors:** Weiqi Wang, Xiangyue Liu, Jesús Pérez-Ríos

**Affiliations:** 1Fritz-Haber-Institut der Max-Planck-Gesellschaft, Faradayweg 4-6, 14195 Berlin, Germany; wang@fhi-berlin.mpg.de (W.W.); xyliu@fhi-berlin.mpg.de (X.L.); 2Department of Physics and Astronomy, Stony Brook University, Stony Brook, NY 11794-3800, USA; 3Institute for Advanced Computational Science, Stony Brook University, Stony Brook, NY 11794-3800, USA

**Keywords:** reaction dynamics, buffer gas chemistry, reaction mechanism

## Abstract

AlF is a relevant molecule in astrochemistry as a tracer of F-bearing molecules. Additionally, AlF presents diagonal Franck-Condon factors and can be created very efficiently in the lab, which makes it a prototypical molecular for laser cooling. However, very little is known about the reaction dynamics of AlF. In this work, we report on the reaction dynamics of AlF–AlF between 200 and 1000 K using *ab initio* molecular dynamics and a highly efficient active learning approach for the potential energy surface, including all degrees of freedom. As a result, we identify the main reaction mechanisms and the lifetime of the intermediate complex AlF–AlF relevant to astrochemistry environments and regions in buffer gas cells.

## 1. Introduction

Aluminum monofluoride (AlF) is a closed-shell molecule, showing a ${}^{1}\Sigma$ electronic ground state and a binding energy of approximately 7 eV. AlF shows an extremely diagonal Franck–Condon factor of 0.995 [1] between the ground vibrational states of the X${}^{1}\Sigma$ and A${}^{1}\Pi$ electronic states. This large Franck–Condon factor can be rationalized in light of the sizeable ionic character of the AlF bond, localizing the electron in one of the atoms and acting locally as a cycling center [2]. Furthermore, AlF can be created in buffer gas cells by ablating a solid Al target in the presence of a fluorine-donor gas. This leads to a chemical reaction that produces AlF as the main product. This chemical reaction is very efficient, yielding a large density of molecules, surpassing any other diatomic molecule explored under the same physical conditions. The high efficiency for AlF formation is related to the large binding energy of AlF of ≈7 eV and reaction mechanisms that minimize the formation of byproducts. In contrast, in other molecules, that is not the case [3].

The efficient production of AlF in buffer gas cells and their almost diagonal Franck–Condon factors make AlF an excellent candidate for laser cooling, thus opening the possibility of reaching the ultracold regime. In that scenario, due to the polar nature of AlF, it could be used for the design of quantum information gates [4,5,6] and the simulation of quantum many-body Hamiltonians [7]. However, other molecules, like bi-alkali molecules, can serve the same purpose. These molecules, in the ultracold regime, show long-lived complexes known as sticky collisions [8,9,10,11,12,13]. Those complexes induce trap losses via photodissociation through the trapping laser or the absence of a proper optical potential to trap them, as has been experimentally corroborated [13,14,15,16,17]. In some bi-alkali systems, discrepancies between the predictions and observations are found for the lifetime of the complex [15,16,17]. These losses compromise most of the applications of ultracold molecules, and they must be controlled. One way to protect bi-alkali molecules against unwanted losses is via microwave shielding [18,19]. However, molecules showing short-lifetime complexes will not require further tools to avoid losses.

On the other hand, AlF is of interest in astrochemistry. AlF appears in the envelope of carbon stars [20,21,22], proto-planetary-nebulae [23], and it has been recently detected in M-type asymptotic giant branch stars [24]. Moreover, ^26^Al^19^F has been the first radioactive molecule ever detected in the remnant of a stellar merger [25]. In all these scenarios, AlF is one of the main two carriers of ^19^F, thus serving to trace the amount of this element whose cosmic origin is still under debate [26]. Also, since AlF contains a metal atom, accounting for refractive materials in the gas phase rather than being condensed onto dust grains. Hence, understanding the chemistry of AlF is required to explain the abundance of this molecule in the different regions observed.

Despite the remarkable properties of AlF, its prospects for laser cooling, and its possibilities for quantum information sciences, quantum simulations, and astrochemistry, very little is known about the chemistry of AlF, except for electron-molecule [27], photo-dissociation dynamics [28], AlF-para-H_2_ rotational inelastic cross section calculations [29], rotational quenching processes for AlF in the presence of a helium buffer gas [30], and a recently developed full-dimensional AlF–AlF potential energy surface (PES) [31]. Therefore, a more detailed study of the chemistry of AlF is required to elucidate the unique properties of this molecule.

In this work, we study via *ab initio* molecular dynamics (AIMD) simulations the reaction dynamics of AlF–AlF between 200 and 1000 K, relevant in astrochemistry and AlF dynamics in buffer gas cells. We infer the lifetime of the intermediate four-body complex Al_2_F_2_ using long-time propagation dynamics within a spherical volume. Alongside this study, we provide reaction models, identifying the most critical configurations in the reaction pathway for inelastic scattering, which is the dominant reaction process at the temperatures under consideration.

Normally, a full-dimensional calculation is only achievable at low-level electronic structure theory using a relatively small basis set. However, using the recently constructed PES for AlF–AlF [31] within an active learning scheme, and combining it with the replica-exchange molecular dynamics (REMD) enhanced sampling algorithm [32,33], we show that it is possible to efficiently use AIMD simulations at coupled-cluster theory with single, double, and perturbative triples excitations [CCSD(T)] level of theory, using a large basis set [31]. In particular, by utilizing the active learning scheme [34,35,36,37,38,39,40,41], on-the-fly AIMD simulations have been performed, commencing with an initial training set consisting of configurations sampled from AIMD at different temperatures. Despite the diverse configurations initially present in the training set across different regions of the PES, the dynamic nature of AIMD simulations at finite temperatures can lead to extensive exploration of regions dissimilar to the ones in the training set, thereby compromising accuracy compared to interpolation regions. Consequently, including CCSD(T) calculations becomes imperative, albeit their computational expense. Therefore, it is efficient to selectively conduct CCSD(T) calculations solely for configurations that display prediction uncertainties surpassing a specific criterion. In the meantime, REMD can effectively accelerate the convergence of the configurational space sampling, reducing the required computational time. As a result, it becomes feasible to construct highly accurate PES for AlF–AlF with a short simulation time and less than 0.01% of configurations requiring CCSD(T) calculations. The resulting PES has demonstrated high accuracy and successfully replicated the stable geometry and energies obtained through CCSD(T).

## 2. Results and Discussions

The MD simulations for the AlF–AlF system are constrained to a sphere of radius $r_{\mathrm{wall}}$ = 8 Å. This region is further divided by introducing a cutoff $R_{\mathrm{cutoff}}$ = 4 Å, such that the complex forming region (region I) appears for $R<R_{\mathrm{cutoff}}$. In contrast, the collision-free region (two non-interacting AlF molecules, region II) is when $R>R_{\mathrm{cutoff}}$, as schematically presented in Figure 1. Region I is prone to forming a four-body complex as represented in Figure 1. This is the region of interest to identify the presence of intermediate complexes. On the contrary, region II is characterized by free motion between the monomers and its elastic collision from the sphere’s boundary, giving rise to the intrinsic timescale of the system denoted as $\tau_{I}$.

During the MD simulations, we sample each configuration’s potential energy relative to the four-body complex’s dissociation threshold as a temperature function, thus better understanding the most visited interaction energies and configurations. The results are shown in Figure 2. The figure shows, as expected, that trajectories at higher temperatures visit shorter distances related to the repulsive wall of the dimer. Similarly, low-temperature trajectories populate more significantly low energy states of the dimer. The binding energy of AlF is 6.89 eV [42]. Therefore, for temperatures between 200 and 1000 K, the collision between two AlF molecules should be highly non-reactive since the collision energy is insufficient to open up that reaction channel, as Figure 2 corroborates.

After analyzing the results of our REMD simulation, we can characterize the complex lifetime. During an AlF–AlF collision, an intermediate AlF–AlF complex may appear. When two AlF molecules get closer, they form a transient four-body state due to bound states in the dimer. The lifetime of these complexes, thus, depends on the density of dimer bound states. Similarly, it is possible to infer the lifetime of these intermediates using a time-dependent approach. Our goal is to characterize the lifetime of these complexes and their impact on the AlF–AlF reaction dynamics.

In our simulations, we assume thermal equilibrium. Hence, we work in the canonical ensemble, in which molecules collide with different kinetic energies following the Maxwell–Boltzmann distribution. Furthermore, considering the ergodic hypothesis, it is possible to show that the reaction rate of a given system is the same whether we run many trajectories for a short time or a single trajectory for a long time. The ergodic hypothesis is the essence of the Rice–Ramsperger–Kassel–Marcus (RRKM) theory, which translates into the Arrhenius equation for the reaction rate.

Here, we chose to run a single trajectory for a long time so that the atoms can sample as much of the available phase space as possible. During this trajectory, we look into the formation of complexes and identify their lifetime, i.e., we calculate how long the complexes live after their formation. Following the RRKM theory, the probability to observe an AlF–AlF complex *p* with lifetime τ is given by [10]
(1)$$p=e^{-\frac{t}{\tau }},$$
where *t* denotes the time. Essentially, this function is a time-correlation function that represents the observable’s behavior with time. Consequently, the probability *p* can be computed using a time-correlation function from trajectories as well, allowing the determination of the lifetime τ at each temperature by fitting the equation with the known values of *p* and *t*.

The formation of an AlF–AlF complex follows its dissociation into two AlF molecules via an unimolecular process characterized by an apparent reaction rate *k* given by the Arrhenius Equation (Equation 2)
(2)$$k=Ae^{-\frac{E_{a}}{k_{B}T}},$$
where $E_{a}$ is the apparent activation energy of the complex, *T* is the temperature of the reaction, and *A* is a prefactor that can be considered as the reaction rate constant at infinite high temperatures. In this scenario, the lifetime τ can be estimated as the inverse of the dissociation reaction rate of the complex into molecules as
(3)$$\tau =\tau_{0}e^{\frac{E_{a}}{k_{B}T}}$$
where $\tau_{0}$ is the inverse of the prefactor *A*. Finally, comparing the results from MD simulations at different temperatures with Equation (Equation 3), $\tau_{0}$ and the activation energy $E_{a}$ can be obtained. However, the presence of the spherical barrier introduces an intrinsic time scale $\tau_{I}$ in our system, related to the time that the molecules take to reach the wall and come back to the scattering center, and hence Equation (Equation 3) has to be modified as
(4)$$\tau =\tau_{0}e^{\frac{E_{a}}{k_{B}T}}+\tau_{I}.$$
In the simulations, the identification of the complex relies only on the distance between two AlF molecules. Here, the lifetime of the AlF–AlF complex is calculated via a REMD simulation based on the PES model explained in Section 3.2.

As a result, we have calculated the lifetime of the complex at different temperatures, and the results are listed in Table 1. We observed that the lower the temperature, the larger the lifetime is, as expected. Indeed, a more detailed study on these numbers shows that the lifetime and reaction temperature follow an exponential–inverse relationship, as anticipated in Equation (Equation 3) and displayed in Figure 3. Hence, AlF–AlF scattering is ergodic, and the lifetime of the complex can be described through the RRKM theory. In other words, we can fit our numerical results to Equation (Equation 4), finding $\tau_{0}=0.0182\left(109\right)$ fs and $E_{a}=0.3356\left(142\right)$ eV, and an off-set time scale $\tau_{I}=$0.0246(56) ps.

During the REMD simulation, we calculate the probability of forming AlF–AlF tetra-atomic complexes, finding the two molecules in a non-interacting stage, as shown in Figure 4. The results indicate that the molecules are mostly part of a complex at low temperatures. However, as the temperature rises, the probability of forming a complex reduces to a constant value of 10%. In other words, the complex formation mainly dominates the reaction dynamics at low temperatures. In contrast, the reaction rates at high temperatures are unaffected by those complexes. Precisely, we observe that between 300 and 500 K, the probability of forming a complex reduces drastically, indicating the presence of a threshold temperature. We find that at 400 K, the likelihood of finding the system in a complex or monomer configuration is the same, thus defining a possible threshold temperature for the role of intermediate complexes in AlF–AlF reaction dynamics.

Last, we have undertaken a clustering approach on the MD-sampled configurations to gain deeper insights into the processes behind AlF–AlF reaction dynamics. These configurations, characterized by their structural similarities, have been grouped into 19 clusters, each denoted by a corresponding landmark. The landmarks are determined based on the high-dimensional Cartesian distances between configurations in the representation space. The methodology for constructing reaction pathways has been elaborated in our previous work [43]. We have adopted the same structural representations detailed in [31]. Subsequently, leveraging these clusters, we constructed Markov state models using the PyEMMA package [44], and inferred reaction pathways through the application of transition path theory [45,46]. In the reaction pathways, the energy associated with a cluster is determined by calculating the average energy derived from all configurations within that cluster.

As a result, we analyze the transitions between distinct configurations of the AlF dimer as a function of the temperature, as depicted in Figure 5. This figure shows the transient configurations during the formation of the AlF–AlF complex, which is the configuration with the lowest energy. These configurations are the most relevant steps of the reaction dynamics. With increasing temperature, more unstable configurations become evident. However, irrespective of the orientation of AlF, there are no barriers during these transitions. This phenomenon has also been discussed in [31], where it is noted that no transitional or intermediary states can be located on the PES, and only a single stable AlF–AlF complex configuration exists contrary to other metal–fluorine molecules like CaF, showing different intermediate states [47].

At 200 K, the complex can form directly from an initially anti-parallel configuration. In this case, there is no barrier to any molecular orientation, facilitating an effortless transition to the stable configuration. Considering that two AlF molecules always exhibit attraction in an anti-parallel configuration, it is reasonable that the direct pathway always contributes fifty percent in the complex formation, spanning from low to high temperatures. Nevertheless, at higher temperatures, we do observe that the orientation of AlF molecules leads to unstable configurations with mutual repulsion, as shown in Figure 5b,c. These new configurations open up new reaction pathways between the reactants and the complex, leading to a more involved reaction pathway network. Similarly, the presence of more possibilities weakens the direct reactant–complex connection, linked to our results presented in Figure 4 on the probability of observing dimer states versus monomer ones.

## 3. Theory and Methods

The AlF–AlF collision dynamics are investigated using AIMD simulations in a wide range of temperatures: 200–1000 K. The underlying PES is calculated using machine learning, making on-the-fly calculations computationally inexpensive. The input for the PES is based on high-level *ab initio* electronic structure calculations, CCSD(T), using the aug-cc-pVQZ basis set. With this approach, it is possible to run MD simulations efficiently to temperatures as low as 200 K.

### 3.1. Molecular Dynamics Simulations

Our MD simulations rely on the velocity Verlet algorithm [48] to integrate Newton’s equation of motion with a time step Δt of 2 fs. All the trajectories are computed within a canonical ensemble (NVT) using the stochastic velocity rescaling algorithm [49]. In this approach, the system is coupled to a heat bath characterized by a parameter of the thermostat of 10 fs. The *ab initio* force is computed using the second-order Møller–Plesset perturbation theory (MP2) with the aug-cc-pVQZ basis set [50]. MP2 is known for being slightly less accurate than CCSD(T) in estimating energy gradients [51,52] but considerably computationally cheaper; therefore it has been used in the construction of machine-learning PES with CCSD(T)-accuracy energies [51,53]. Furthermore, Molpro provides efficient MP2 analytical gradient calculation, making it a practical choice [54]. Therefore, this represents the highest level of analytical gradients currently computationally affordable for an MD simulation at the aug-cc-pVQZ level for the AlF dimer system.

Instead of simulating many initial configurations for a short time, we use the ergodic hypothesis to run one trajectory for a long time. However, we constrain the motion of the dimer into a spherical volume to ensure that many collisions occur during the trajectory. We use a repulsive spherical wall with 8 Å radius to simulate the spherical volume. This radius guarantees the molecules enough time to relax between two consecutive encounters. The potential energy of the repulsive wall is modeled following a power law as
(5)$$U_{wall}=\left\{\begin{array}{cc} a\cdot \left|\right|\vec{b}{\left|\right|}^{c} & \left(r_{atom}⩾r_{wall}\right)\\ 0 & \left(r_{atom}⩽r_{wall}\right) \end{array}\right.$$
while the forces acting on the atoms from the wall are
(6)$${\vec{F}}_{wall}=\left\{\begin{array}{cc} a\cdot c\cdot \left|\right|\vec{b}{\left|\right|}^{c-1}\frac{\vec{b}}{\left|\right|\vec{b}\left|\right|} & \left(r_{atom}⩾r_{wall}\right)\\ 0 & \left(r_{atom}⩽r_{wall}\right) \end{array}\right.$$
*a* (a=1) defines the scaling of the wall potential, $\vec{b}$ is the normal vector starting from the atom and ending at the wall, *c* (c=2) defines the power of the repulsive potential of the wall, $r_{atom}$ is the distance between the sphere center and the atom, and $r_{wall}$ is the radius of the sphere. This form of spherical potential ensures that the action range is finite and adjustable, and the potential energy surface in the whole space of the system is differentiable in the entire space. As a result, the two AlF molecules will collide multiple times, forming intermediate four-body complexes.

To optimize the simulation time required to approach the dissociation equilibrium at low temperatures, an enhanced sampling method, REMD, has been employed [32,33]. In the REMD method, the system is simulated at a series of different temperatures using a certain number of replicas, i.e., copies of the system, that are exchanged periodically. In this framework, simulations at temperatures higher than the physical temperature accelerate the exploration of the configurational space, visiting otherwise unreachable areas. Therefore, the enhanced sampling technique guarantees a fast convergence of the MD sampling towards the thermodynamical equilibrium. The time scales of the simulations can be extended via extrapolation from converged equilibrium results. In particular, we simultaneously initiate multiple trajectories at ten different temperatures (200 K, 239 K, 286 K, 342 K, 408 K, 489 K, 585 K, 699 K, 836 K, and 1000 K), which are then swapped during the simulation after a certain number of steps. The exchange process follows the Boltzmann probability distribution and satisfies the detailed balance condition. This approach facilitates a faster thermodynamic equilibrium with the aid of high-temperature replicas. The total simulation time amounts to 5.4 ns in each of the ten replicas.

### 3.2. Computational Details

Based on the PES constructed in Ref. [31], we have performed a REMD simulation with the aid of an active learning scheme described in Ref. [31] to make our computations more efficient. To ensure the reliability of our machine-learning-based PES model, we have calculated *ab initio* energies at the CCSD(T) level of theory as implemented in the Molpro package [54]. During AIMD simulations, the *ab initio* forces are calculated at the MP2 level. The calculations were performed with the aug-cc-pVQZ basis set [50,55,56].

The PES is constructed using a two-body structural representation composed of inverses and exponentials of interatomic distances. It preserves essential invariant symmetries, such as translational, rotational, and permutational invariance, thus satisfying the exchange symmetry of the system at hand. Therefore, these features are well suited for any machine learning approach to characterizing a tetra-atomic system’s energy landscape. The list of structural information, including its ab inito energy, is divided between training and test sets. The training set contains the configurations and energies exposed to the machine learning algorithm to learn the relationship between configuration and energy. On the contrary, the test set contains new configurations that have yet to be shown to the algorithm used to test the efficiency of the learning procedure. Ideally, we would like to use the smallest training set possible, leading to the most accurate prediction or efficient learning. Here, we use a variant of this approach called active learning, in which the training set is enlarged depending on the precision of a prediction and target accuracy. Specifically, the machine learning algorithm is used to infer the energy of a new configuration required for the MD simulations. Suppose the uncertainty of this prediction is larger than a threshold value. In this case, this configuration is calculated via *ab initio* quantum chemistry methods, and the data point is included in the training set. Otherwise, the procedure continues with the same training set.

The accuracy of the fitted PES has been demonstrated in Ref. [31]. Nonetheless, we provide a brief overview here. During the MD simulation of the AlF–AlF system, the enhanced-sampling REMD simulation uses an active-learning approach, i.e., the training size set grows, as required, during the simulation to guarantee a given accuracy of the energy predictions. Following this approach, we have shown that with only 10,000 *ab initio* points, it is possible to accurately predict the outcome of an MD trajectory consisting of 3633 steps with a mean absolute error of 0.85 meV/atom (or a median absolute error of 0.019 meV/atom).

In this work, the initial training set counts on 22,365 CCSD(T)/aug-cc-pVQZ energies sampled by AIMD at different temperatures to ensure a proper sampling of the configuration space. Incorporating additional *ab initio* data during on-the-fly simulations through the active-learning approach further enhances the description of the PES in regions not covered in the initial training set. This has proven particularly crucial for repulsive regions characterized by short intermolecular distances and high energies, as these regions can be heavily visited during the MD simulations. Using the active-learning/REMD simulation approach, an additional 2038 configurations (∼0.008% of the REMD-sampled configurations) are calculated *ab initio*, selected based on prediction uncertainty. The final PES can precisely reproduce the CCSD(T)/aug-cc-pVQZ-optimized geometry of the only stable configuration of AlF dimer, with differences in Al-F bond lengths of less than 0.0004 Å. Moreover, the predicted total energy is only 2.5 meV higher than the CCSD(T)/aug-cc-pVQZ result.

## 4. Conclusions

In this work, we studied the AlF dimer’s reaction dynamics between 200 and 1000 K via replica-exchange *ab initio* molecular dynamics simulation, employing a highly accurate full-dimensional potential energy surface. The calculations are performed on the fly, with an active learning approach, offering a highly accurate description of the potential energy landscape and the dynamics in an affordable computational time. Within this theoretical framework, in virtue of the ergodic hypothesis, we ran parallel tempering trajectories, constraining the AlF dimer dynamics in a spherical volume. As a result, apart from fully characterizing the reaction dynamics, we extracted the lifetime of the single AlF dimer complex as a function of the temperature, and analyzed its impact on the reaction dynamics.

The AlF dimer reaction dynamics shows an ergodic behavior since it follows the Arrhenius law. Consequently, the lifetime of the complex can be explained by the RRKM theory for unimolecular reactions. The reaction dynamics only show inelastic events rather than reactive ones, as expected due to the large binding energy of AlF. The probability of observing complexes during a reaction depends on the temperature, showing a drastic change at temperatures ∼400 K. At temperatures below 400 K, the probability to form an intermediate complex is almost 100%, significantly impacting the reaction dynamics. On the contrary, at temperatures higher than 400 K, the reaction dynamics is virtually unaffected by the intermediate complex, and the probability of forming a complex drops to 10%. This behavior is corroborated by our reaction models as a function of the temperature based on Markov state models.

Finally, the non-reactive nature of the AlF dimer indicates that AlF will be highly stable in carbon-rich stellar atmospheres, only showing vibrational and rotational excitations. However, the explicit role of these has yet to be studied. Similarly, it would be necessary to study the reaction dynamics at temperatures ≲200 K for the case of proto-planetary regions. Nevertheless, based on our reaction models, the intermediate complex will play a role in inelastic transitions, and they will become more relevant as the temperature drops. In the case of buffer gas cells, our results indicate that once AlF appears after the Al–fluorine-donor reaction, it will be stable, only experiencing inelastic transitions after colliding with the He buffer gas, guaranteeing a cold and intense beam of AlF molecules after leaving the buffer gas cell.

## Figures and Tables

**Figure 1 molecules-29-00222-f001:**
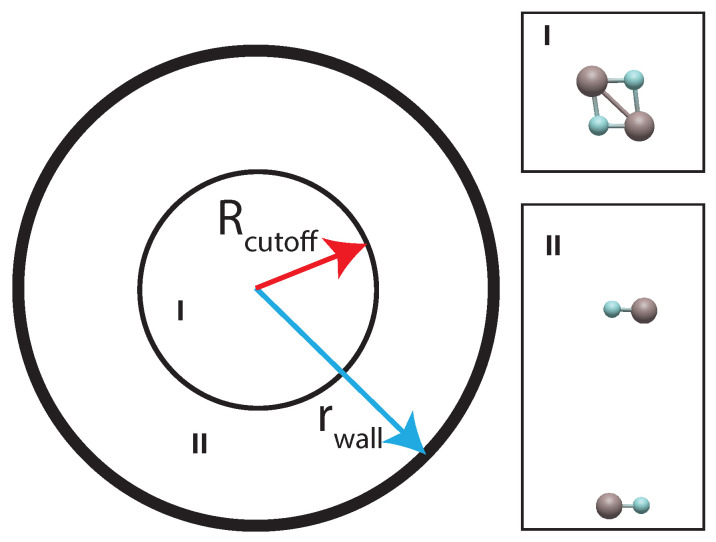
Potential for complex characterization. Regions I and II refer to the complex forming and collision-free regions, respectively. See text for details.

**Figure 2 molecules-29-00222-f002:**
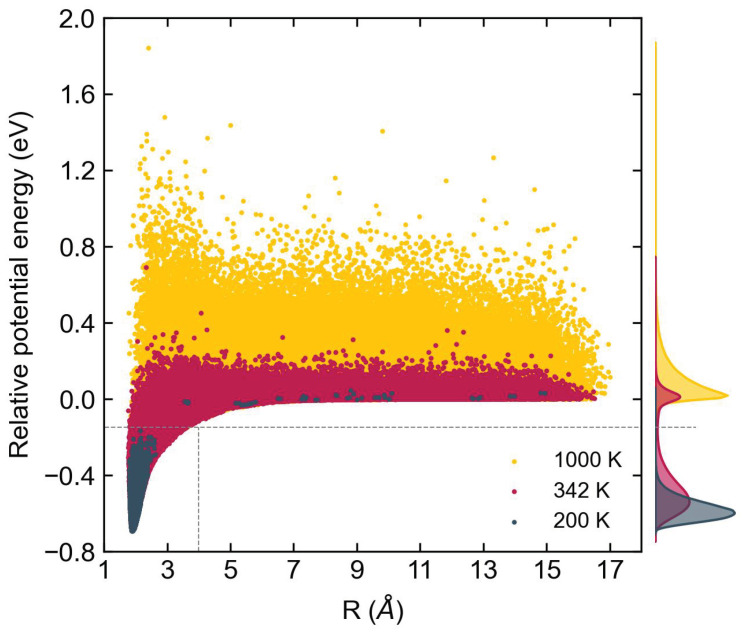
Potential energy distribution. Potential energy distribution of the sampled configurations during the simulation of AlF–AlF reaction dynamics. The energies are referenced to the energy of dissociated AlF–AlF complex with intermolecular distance *R* = 20 Å. The distribution at 200 K, 342 K and 1000 K are shown in grey, red and yellow, respectively.

**Figure 3 molecules-29-00222-f003:**
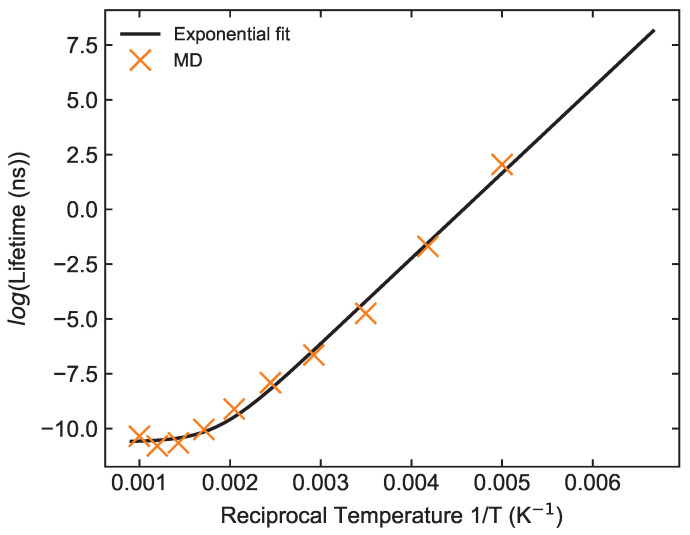
The lifetimes at different temperatures are fitted by Equation (Equation 4). The parameters $\tau_{0}$, $E_{a}$, and $\tau_{I}$ are obtained as 0.0182(109) fs, 0.3356(142) eV, and 0.0246(56) ps, respectively.

**Figure 4 molecules-29-00222-f004:**
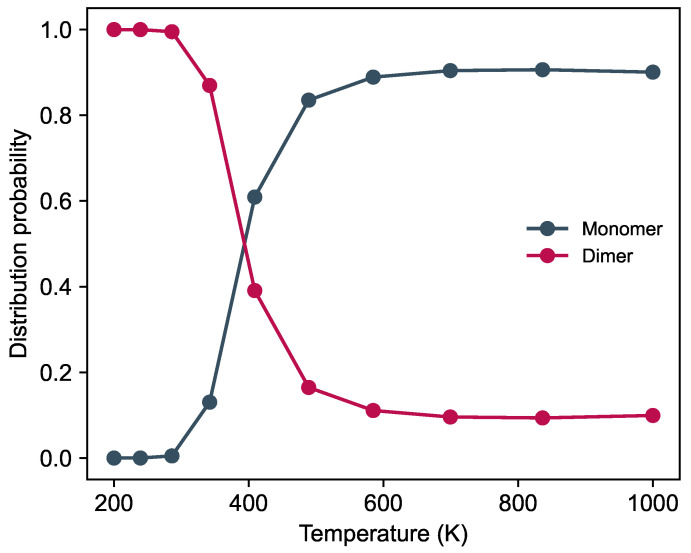
Distribution probability of AlF–AlF dimer complex and the dissociated monomers, as a function of temperature.

**Figure 5 molecules-29-00222-f005:**
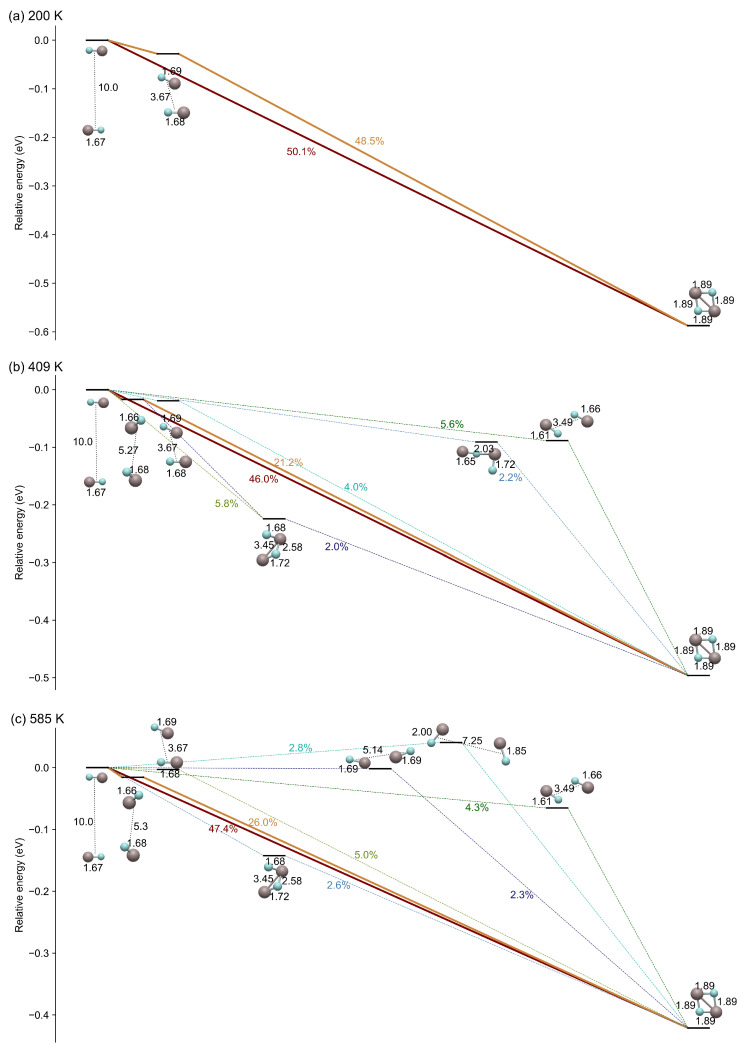
Transition pathways at different temperatures. On the left side are two separated AlF molecules, located at a minimum distance of 10 Å. On the right side is the AlF–AlF complex, representing the lowest potential energy point in PES. The molecules located between these two configurations are intermediates involved in the formation of the AlF–AlF complex. The percentage numbers indicate the respective contributions of reaction paths labeled by colors. The black numbers indicate the intermolecular distances and the Al-F bond lengths in Å.

**Table 1 molecules-29-00222-t001:** The lifetimes of AlF–AlF dimer at different temperatures.

Temperature (K)	Lifetime (ps)
200	7758.1(6)
239	185(10)
286	8.7(1)
342	1.30(2)
408	0.37(1)
489	0.111(5)
585	0.043(2)
699	0.024(1)
836	0.0205(6)
1000	0.032(1)

## Data Availability

The data presented in this study are available on request from the corresponding author.

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
