# Peer review of "AlF–AlF Reaction Dynamics between 200 K and 1000 K: Reaction Mechanisms and Intermediate Complex Characterization"

_molecules, 2023, doi:10.3390/molecules29010222_

Round 1

Reviewer 1 Report

Comments and Suggestions for Authors

I thank the authors for a job well done. I have no important comments on the manuscript. However, I would like to note two points:

1. Line 126 - instead of a link in square brackets there is a question mark. Needs to be corrected. 

"The binding energy of AlF is 6.89 eV [?  ]."

2. This is a proposal to the authors. It might make sense to provide the geometric parameters of the structures - intermediates - presented in Figure 5. This may clutter up the manuscript, but this can easily be done into separating the Supplemental Material file

Reviewer 2 Report

Comments and Suggestions for Authors

AlF is a related molecule in astrochemistry, serving as a tracer for fluorinated molecules and a typical molecule for laser cooling. Therefore, studying its dynamics is crucial. The manuscript combines ab initio molecular dynamics with efficient machine learning methods to construct potential energy surfaces. Based on this, the reaction kinetics of AlF-AlF between 200 and 1000K were studied, and the main reaction mechanism and lifetime of the intermediate complex AlF-AlF were determined. The research is very meaningful and interesting. The overall writing of the manuscript is good. However, there are the following issues that need further improvement and enhancement in revised manuscript

1. Lack of effective experimental results to ensure the accuracy of the constructed potential energy surface. Simply says, the author should provide the reliability of the methods used. Information such as whether the system is converging has to be provided.

2. The comparison and discussion between the results of CCSD (T) and MP2 calculations and experimental results were not provided.

3. The author needs to carefully check the manuscript and eliminate many unclear contents. Such as "The binding energy of AlF is 6.89 eV [?]." The authors noted that "The orientation of AlF molecules leads to unstable configurations with multiple replication, as shown in panels (b) and (c) of Fig. 5.", but Figure 5 is not given in the manuscript?

Reviewer 3 Report

Comments and Suggestions for Authors

The manuscript “AlF-AlF reaction dynamics between 200K and 1000K: reaction mechanisms and intermediate complex characterization” by Weiqi Wang, Xiangyue liu and Jesús Pérez-Ríos discuss a molecular system which is of particular interest for astrochemistry environments and regions in buffer gas cells. The authors give an exhaustive introduction and detailed methodology section. The results and discussion are interesting and well presented.

Minor corrections are requested, and few suggestions are proposed to make this work more conclusive.

Line 40-41: Please correct the sentence “…–the dominant reaction process at the temperatures under consideration for different temperatures.”

Line 75: The authors write that “total simulation time amounts to 5.4 ns in each of the ten replicas.” Then in tab 1 at 200K the lifetime is 7.758 ns. Could you give details concerning how can you calculate a lifetime greater than the simulation time? Maybe additional details are needed in the “Theory and Methods” paragraph.

Eq. 5: Comment on the implicit hypothesis you have adopted about A as constant with temperature. Why have you not used a modified Arrhenius equation?

Line 127: The authors write: “...the collision between two AlF molecules should be highly non-reactive. This is because it is very unlikely that all the kinetic energy is deposited in the internal degree of freedom of the molecule, suggesting a small coupling between the internal degrees of freedom and the kinetic ones.”. I suggest to the authors that the collisions between two AlF molecules is not reactive at 1000 K not because is very unlikely that all the kinetic energy is deposited in the internal degree of freedom of the molecule - which the authors should demonstrate and which is not obvious, but because by simply applying the Boltzmann distribution only about 2x10^-35 molecular fraction has the minimum amount of energy (6.89 eV) to break the AlF molecule. The key point is not the vibro-translational coupling, but just the standard thermal kinetic distribution which prevents the breaking of AlF at 1000K and at lower temperatures.

The authors affirm in the abstract that “...we identify the main reaction mechanisms and the lifetime of the intermediate complex AlF-AlF relevant to astrochemistry environments and regions in buffer gas cells.” This statement should be discussed in the paragraph 3 in relation with the number density of AlF used by the authors throughout all their simulations. The authors have used a spherical wall with 8 Å radius, which implies a large number density of AlF. This “high” number density should be related with those present in the astrochemistry environments. Otherwise, if the comparison between the number density used by the authors and that of the astrochemistry environments is meaningless, I suggest to the authors to modify their statement that directly relate the calculated lifetimes with those of the astrochemistry environments.

Round 2

Reviewer 2 Report

Comments and Suggestions for Authors

The authors have revised the manuscript acoording to all my suggestion.The revised manuscript can  be accepted as it is.

Author Response

Thank you for your thorough review and constructive suggestions.